# Integrating Suboptimal Human Knowledge with Hierarchical Reinforcement Learning for Large-Scale Multiagent Systems

**Dingbang Liu[1,2], Shohei Kato[2], Wen Gu[3], Fenghui Ren[1], Jun Yan[1], Guoxin Su[1]**
University of Wollongong[1], Nagoya Institute of Technology[2],
Japan Advanced Institute of Science and Technology[3]
dl956@uowmail.edu.au, shohey@nitech.ac.jp, wgu@jaist.ac.jp
fren@uow.edu.au, jyan@uow.edu.au, guoxin@uow.edu.au

## Abstract

Due to the exponential growth of agent interactions and the curse of dimensionality, learning efficient coordination from scratch is inherently challenging in large-scale multi-agent systems. While agents' learning is data-driven, sampling from millions of steps, human learning processes are quite different. Inspired by the concept of Human-on-the-Loop and the daily human hierarchical control, we propose a novel knowledge-guided multi-agent reinforcement learning framework (hhk-MARL), which combines human abstract knowledge with hierarchical reinforcement learning to address the learning difficulties among a large number of agents. In this work, fuzzy logic is applied to represent human suboptimal knowledge, and agents are allowed to freely decide how to leverage the proposed prior knowledge. Additionally, a graph-based group controller is built to enhance agent coordination. The proposed framework is end-to-end and compatible with various existing algorithms. We conduct experiments in challenging domains of the StarCraft Multi-agent Challenge combined with three famous algorithms: IQL, QMIX, and Qatten. The results show that our approach can greatly accelerate the training process and improve the final performance, even based on low-performance human prior knowledge.

## 1 Introduction

As an essential attribute of multi-agent reinforcement learning (MARL), scalability deserves more attention since it contributes to autonomous collective learning behaviors among agents. In many engineering and scientific disciplines, algorithms must possess sufficient scalability to cooperate properly [1, 2, 3]. Unfortunately, training a large number of agents presents inherent challenges. The joint action-state space increases exponentially with the number of agents, resulting in the curse of dimensionality [4, 5]. Agents, suffering from sparse rewards and sample inefficiency [6, 7], encounter 'start-up' problems [8] and often get trapped in local optima, to the extent that learning becomes impossible in the worst cases [9, 10].

To alleviate the exploration burden and overcome the curse of dimensionality, knowledge transfer methods have attracted significant research interest in dealing with large-scale multi-agent systems (MAS) [11, 12, 13]. As the most intuitive and common source, transferring prior knowledge from humans has received considerable attention [14, 15]. An important line of research involves extracting prior knowledge from expert trajectories through imitation learning to address sequential decision-making problems [16, 17]. However, despite the emphasis on Human-on-the-Loop [18], most research still focuses on step-by-step action demonstrations, which require high-quality and comprehensive prior knowledge [11, 14, 17]. The deployment of this type of human knowledge becomes increasingly

38th Conference on Neural Information Processing Systems (NeurIPS 2024).

difficult in complex tasks, necessitating overwhelming human effort. Additionally, creating human-guided state-action pairs for each scenario is so time-consuming that even human experts struggle to anticipate actions in these highly challenging tasks.

Fortunately, human guidance is not limited to step-by-step action demonstrations, and other high-level knowledge exists to reduce the human effort involved [14]. Human prior knowledge can be abstracted and transferred to agents through abstract rules or heuristics [19, 20, 21]. Although the provided knowledge is suboptimal, it can greatly reduce the computational burden and improve algorithm performance. However, current discussions are still limited to single- or two-player scenarios. Unfortunately, utilizing human prior knowledge properly is significantly more challenging in MAS. Due to non-stationarity and multiple Nash equilibria, most single-agent methods are unsuitable for multi-agent scenarios. Additionally, human knowledge is usually applied indiscriminately, and mapping knowledge from humans to agents is still in its early stages [15].

To better transfer knowledge from humans, it's essential to understand the nature of human knowledge. Despite accomplishing many complex tasks in daily life, human activities do not require much active attention (Figure 1). When deciding to get up and walk, the brain uses the prefrontal cortex to generate commands, but the intricate coordination of sensory inputs and muscles that follows does not require any conscious attention. Instead, it's mostly executed by a lower nerve network in the spinal cord, sometimes called central pattern generators (CPG) [22]. Indeed, humans are naturally adept at abstracting and providing high-level knowledge, and

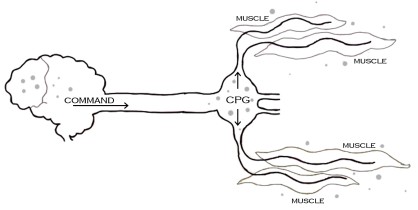

Figure 1: Human daily hierarchical control.

asking them to provide detailed step-by-step action guides is overly demanding. It may be preferable to have humans deliver abstract knowledge at the top level, while agents spontaneously decide on the utilization of the proposed knowledge based on their 'CPG'.

Inspired by the hierarchical control of human daily activity, we propose a novel knowledge-guided MARL framework to integrate abstract human knowledge into MARL algorithms in an end-to-end manner. As several works have demonstrated, a hierarchical structure [23] and graphical models [1] can greatly benefit large-scale MAS. We believe our approach, the hierarchical human knowledge multi-agent reinforcement learning framework (hhk-MARL), can alleviate learning difficulties among a large number of agents. In our framework, trainable fuzzy logic rules are applied to represent human knowledge. Since the prior knowledge is suboptimal, our framework allows agents to adjust the utilization of proposed knowledge through hyper-network based knowledge integration. Additionally, to enhance agent coordination, we construct a graph to determine the importance of agents and simplify their relationships. By applying human knowledge at the top level while maintaining agents' ability to develop self-policies at the bottom level, our hierarchical method helps 'warm-start' the training process and overcomes the learning difficulties of large-scale MAS. Experimental results from challenging tasks in the StarCraft Multi-agent Challenge (SMAC) [24] demonstrate that our approach can be easily combined with various MARL algorithms, significantly improving their learning efficiency.

## 2 Preliminaries

### 2.1 Partially observable Markov game

We view our problem as a cooperative multi-agent task. This task can be modeled as a decentralized partially observable Markov decision process (Dec-POMDP) [25], given by a tuple $G$:

$$G = \langle S, U, P, r, Z, O, n, \gamma \rangle \tag{1}$$

where $s \in S$ defines the global environment state. The observations of each agent $i \in N \equiv \{1, \ldots, n\}$ are partially observable $o^i \in O^i$, which are determined by the observation function $Z(s, i) : S \times N \to p(O)$. Based on its local observation $o_i$, each agent $i$ selects its action $u_i \in U_i$ at each time step according to its stochastic policy $\pi_i : O_i \times U_i \to [0, 1]$. Then, based on the state transition function $P : S \times U_1 \times \cdots \times U_n \to S$, the joint action of all agents $\vec{u} \in \vec{U}$ changes the environment into a new state. All agents share the same reward function $r(s, \vec{u}) : S \times \vec{U} \to R$, and aim to learn poli-

cies to maximize action-value functions $Q(s_t, \vec{u}_t) = \mathbb{E}_{\vec{u}_{t+1} \sim \vec{\pi}, s_{t+1} \sim P}[\sum_{l=0}^{Th-t} \gamma^l r_{t+l}(s_t, \vec{u}_t)|s_t, \vec{u}_t]$ where $\gamma$ is the discount factor and $Th$ is the time horizon.

## 2.2 Fuzzy logic

Since human knowledge is highly abstract and uncertain, it is inappropriate to represent knowledge with hard rules [26]. On the contrary, fuzzy logic can tackle issues of uncertainty and lexical vagueness, pacing its way to depict human imprecise knowledge [20]. A fuzzy logic rule is usually in the form of 'IF $X$ is $A$ and $Y$ is $B$ THEN $Z$ is $C$' with a membership function $\mu$ for each fuzzy set used to calculate the truth value $T$ of each precondition:

$$T_A = \mu_A(x_0) : X \to [0,1], T_B = \mu_B(y_0) : Y \to [0,1] \tag{2}$$

where $x_0$ and $y_0$ are the observation values for $X$ and $Y$, respectively. To derive the conclusion of this fuzzy rule, both preconditions must be satisfied:

$$\mu_{A \cap B}(x_0, y_0) = min(\mu_A(x_0), \mu_B(y_0)) \tag{3}$$

Finally, the conclusion's strength, $\omega$, is obtained as follows:

$$\omega = min(T_A, T_B) = min(\mu_A(x_0), \mu_B(y_0)) \tag{4}$$

Therefore, a fuzzy logic rule takes the observation values as inputs and outputs the value of the conclusion to illustrate how likely it is to operate the designed actions under the current observations.

## 2.3 Knowledge representation and integration

Although human knowledge is highly instructive, such prior knowledge is suboptimal and covers only a small part of the state space. Applying it indiscriminately can hinder the training process. To avoid negative knowledge transfer, agents should have the freedom to decide the utilization of transferred knowledge. For better intuition, a teacher (human) and student (agent) model could be considered. To guide student research, the mentor might give the doctoral student some comments. For example, a comment for literature review could be: *'Read paper with high citation score'*, which can be expressed in the fuzzy logic rule as 'IF $O$ is $high$, THEN $action$ is $read$'. Here, $O$ is the observation of the citation score, and $high$ is a fuzzy set $M$ whose membership function could simply be, $\mu_{high}(o) : clip[0.05o, 0, 1]$. Therefore, the higher the score, the stronger the action $read$ will be. Apparently, such knowledge is abstract and proposed for a specific state in the research process. Indeed, it is unrealistic to require the mentor to design a comprehensive checklist, and uncertainty and lexical vagueness are inevitable. The student is expected to autonomously decide when to follow the mentor's advice and when to trust their own judgment. Inspired by the CPG of human hierarchical control (Figure 1), to integrate abstract human knowledge into the agent learning process, we design a knowledge 'CPG' for the agent to freely decide how to leverage the suboptimal knowledge.

# 3 Human knowledge guided hierarchical framework

In this section, we will illustrate a novel human knowledge-guided MARL hierarchical framework (hhk-MARL) that integrates abstract human knowledge into MARL in an end-to-end manner to enhance agents' learning efficiency. The overall architecture is shown in Figure 2, which is divided into three levels, mimicking the human hierarchical control (Figure 1). Inspired by the CPG of humans, a hyper-network based knowledge integration is built to allow agents elegantly leverage the proposed human prior knowledge. The details of each module will be elaborated in the following sections, and the meanings of symbols can be found in Appendix A.3.

## 3.1 Knowledge controller

Since requiring humans to provide comprehensive step-by-step demonstrations for large-scale MAS is unrealistic, we focus on abstract and suboptimal human knowledge to reduce the human effort involved. As introduced in Section 2.2, fuzzy logic is applied to capture human imprecise knowledge. Compared to other knowledge representation methods, fuzzy logic is closer to the structure of human knowledge, making it more interpretable. Furthermore, it has been proven that fuzzy logic is more suitable for training large-scale multi-agent systems with the advantage of generalization

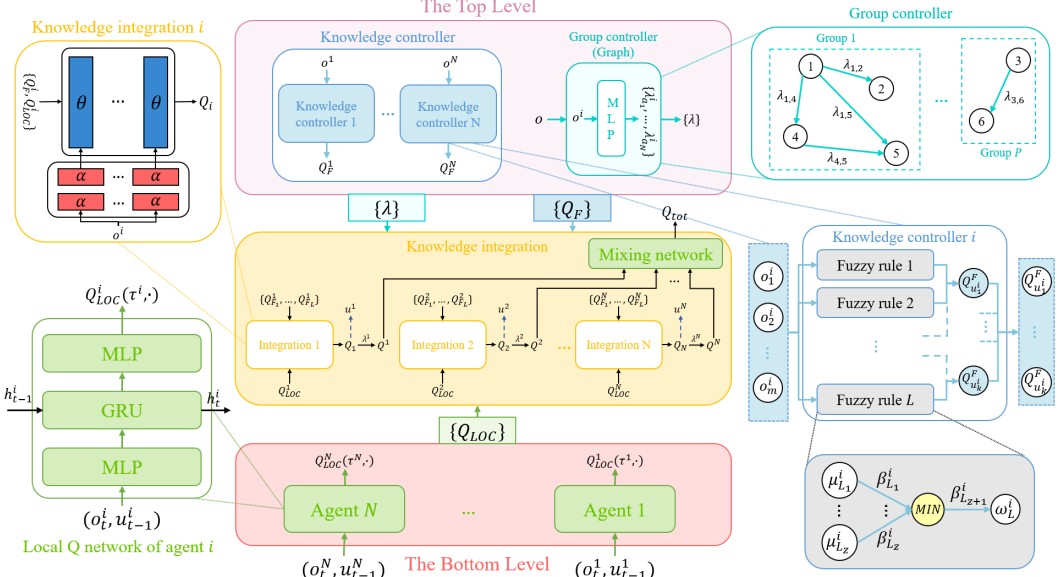

Figure 2: The overall framework of the human knowledge guided hierarchical MARL. The general architecture is proposed in the middle which is separated into three levels. Agents can develop their own policies with traditional MARL algorithm shown at bottom left. The graph-based group controller is depicted at top right to enhance agents' coordination. The knowledge controller is comprised with fuzzy logic rules to represent human knowledge which is demonstrated at bottom right. The hyper-networks of knowledge integration is illustrated at top left to allow agents freely decide the use of proposed human knowledge.

[27]. Inspired by previous works on knowledge representation with fuzzy logic [20, 27], we leverage fuzzy logic to abstract human prior knowledge in this work. The general form of each rule can be represented as:

- *Rule L*: IF $O_1$ is $M_{L_1}$ AND $O_2$ is $M_{L_2}$ AND ... AND $O_z$ is $M_{L_z}$, THEN *Action* is $u_{kk}$

Here, $O_z^i$ are variables about the system extracted from agent $i$'s observations, and $M_{L_z}^i$ are fuzzy sets for the corresponding variables $O_z^i$. The strength of *Rule L* on corresponding actions is obtained:

$$\omega_L = min[\mu_{L_1}(o_1), \mu_{L_2}(o_2), \dots, \mu_{L_z}(o_z)] \tag{5}$$

where $o_z^i$ are observation values for $O_z^i$ and $\mu_L^i$ are membership functions for the fuzzy sets $M_L^i$. An example of fuzzy logic rule design is illustrated in Section 2.3 for clarity. For fairness, humans are only allowed to propose guidance based on agent's local observations.

As the transferred human knowledge is very rough, inspired by [20], we add trainable weights $\beta$ to adapt proposed knowledge to current tasks. With these trainable weights, the knowledge can be optimized similarly to a neural network. For each rule, there are $z + 1$ corresponding weights: the first $z$ weights adjust the causal degree of each precondition, and the last weight $\beta_{z+1}$ indicates the confidence in the proposed knowledge. Therefore, the human preference vector $Q_{F_L}$ for fuzzy rule $L$ on each action can be represented as (an example of human preference vector is in Appendix A.6):

$$Q_{F_L}(u_1, \dots, u_k) = \beta_{L_{z+1}} min[\beta_{L_1} \cdot \mu_{L_1}(o_1), \beta_{L_2} \cdot \mu_{L_2}(o_2), \dots, \beta_{L_z} \cdot \mu_{L_z}(o_z)] \tag{6}$$

These trainable weights are initialized at 1 to avoid disturbing the prior knowledge, and then adjusted through the reinforcement learning based on the reward signal. However, it is worth mentioning that a fuzzy rule can be initialized with a higher weight when there is high confidence in it.

We refer to this knowledge representation module as the knowledge controller, shown in Figure 2 bottom right. To maintain the merit of scalability, this knowledge controller is shared among all agents. Specifically, for agent $i$, the knowledge controller takes the agent $i$'s observation $o^i$ as input

---

**Algorithm 1** Human knowledge guided hierarchical MARL

---

**Input:** MARL algorithm, Knowledge controller, Group controller, Knowledge integration
**Output:** Human knowledge guided MARL model

1: Initialize the parameters of MARL algorithm, knowledge controller, group controller, knowledge integration
2: **for** episode = 1 to max-episode **do**
3:     **for** $t$ = 1 to max-episode-length **do**
4:         **for** agent $i$ = 1 to $N$ **do**
5:             Calculate $Q_{LOC}^i$ based on MARL algorithm
6:             Use fuzzy rules for $Q_F^i = \{Q_{F_1}^i, \ldots, Q_{F_L}^i\}$
7:             Input $\{Q_{LOC}^i, Q_F^i\}$ into knowledge integration for knowledge guided vector $Q_i$
8:             Sample action $u^i$ from $Q_i$ with $\epsilon$-greedy policy
9:         **end for**
10:         Execute $(u^1, \ldots, u^N)$ to obtain reward $r$ and new observation $o^{'}$ from environment
11:         Store $(o, u, r, o^{'})$ in replay buffer $D$, and set $o \leftarrow o^{'}$
12:         Sample a random minibatch of $G$ samples $(o^g, u^g, r^g, o'^g)$
13:         Obtain $\{\lambda^1, \ldots, \lambda^N\}$, set $Q_{tot}$, and update the Q-network by minimizing the loss: $\mathcal{L}_{tot}$
14:     **end for**
15: **end for**

---

and outputs $L$ number of human preference vectors based on built-in fuzzy logic rules:

$$Q_F^i = \left\{Q_{F_1}^i, Q_{F_2}^i, \ldots, Q_{F_L}^i\right\} \tag{7}$$

### 3.2 Knowledge integration

Although trainable weights are added to the knowledge controller to mitigate the knowledge mismatch problem, the proposed knowledge is still quite rough, covering only a small portion of the state space. Since humans and agents have different perceptions and knowledge structures, it is more appropriate to allow agents to determine the utilization of prior knowledge. To not distort human knowledge and avoid negative knowledge transfer, we propose a hyper-networks based knowledge integration to integrate human prior knowledge into agents' policies. Similar to human CPG (Figure 1), this approach allows humans to design abstract knowledge from a high-level, while agents can autonomously decide whether to accept the proposed knowledge and how to leverage it.

Although applying a concatenated neural network as the knowledge integration is straightforward, it is difficult to capture the dynamic knowledge requirements in different states. To allow agents to automatically adapt to human guidance, motivated by previous research [20], we propose a hyper-networks based knowledge integration that allows agents to refine the proposed prior knowledge based on the local observation. As shown in Figure 2 (top left), the knowledge integration consists of two networks that take the human preference vectors $Q_F^i$ and the agent preference vector $Q_{LOC}^i$ as input, and output the knowledge-guided action preference vector $Q_i$ based on the agent $i$'s local observation. Formally, the first network takes the observation $o^i$ of agent $i$ as input and generates weights for the second network, which combines the agent's policy with human knowledge:

$$Q_i = k_\theta(Q_F^i, Q_{LOC}^i) \tag{8}$$

where:

$$\theta = h_\alpha(o^i) \tag{9}$$

Here, $h_\alpha(\cdot)$ is the hyper-network that generates the weights $\theta$ for the integration $k_\theta(\cdot)$. To encourage agents to frequently explore human knowledge at the start, a hyperparameter $\Omega$ is considered:

$$\theta = max(h_\alpha, \Omega) \tag{10}$$

This hyperparameter is initialized to 1 and quickly decreases to zero to not impair the agents' autonomy in knowledge adjustment. Similar to the knowledge controller, the knowledge integration is also trained based on the reinforcement learning process and this module is also shared among all agents.

### 3.3 Group controller

Besides the 'start-up' problem from the curse of dimensionality, another difficulty in large-scale MAS is the intricate interactions among agents, which intensify as the number of agents increases. To enhance agent coordination and further improve scalability, motivated by the graph's ability to simplify relationships [1, 28], a simple neural network is applied to generate a cooperation graph among agents (Figure 2 top right). This group controller uses local observation to output cooperation tendency among agents. Specifically, based on $o^i$, agent $i$ can propose the allies it wants to cooperate with, following the tendency strength $\lambda_{i,j}$:

$$\lambda_i = \{\lambda_{i,1}, \ldots, \lambda_{i,N}\} \tag{11}$$

where $\lambda_{i,j}$ is the cooperation tendency of agent $i$ toward agent $j$. After agents deliver their tendencies, this will form a relationship graph (an example is shown in Figure 9):

$$\{\lambda_1, \ldots, \lambda_N\} \tag{12}$$

Here, an agent's importance is considered based on other allies' cooperation tendencies toward it:

$$\left\{\lambda^1, \ldots, \lambda^N\right\} = softmax(sum(\{\lambda_1, \ldots, \lambda_N\}, dim = -2)) \tag{13}$$

where $\lambda^i$ is the importance weight of agent $i$ and will be used to weight the agent's selected action. The group controller is also shared among all agents. Additionally, to not violate the principle of centralized training with decentralized execution (CTDE), this module will only be applied during the training process.

### 3.4 Knowledge guided hierarchical MARL

To transfer knowledge from humans, we integrate abstract human knowledge into the agent learning process by mimicking the hierarchical control of human daily activities. As demonstrated in Figure 2, human can easily propose abstract knowledge at a high-level, while agents autonomously form specific action-state strategies based on the MARL algorithm. This learning framework is end-to-end and can be combined with various MARL algorithms. To clarify this process, QMIX [29] algorithm is applied as an example here. Initially, the agent $i$'s preference vector $Q^i_{LOC}(\tau^i, \cdot)$ is calculated based on its local Q network. Then, the human preference vectors $Q^i_F(o^i)$ are generated from the knowledge controller, mentioned in Section 3.1. Next, following Equation 8, the human knowledge is integrated into agent $i$'s policy, forming the knowledge-guided action preference vector $Q_i$. This vector $Q_i$ is used to sample the action $u^i$ of agent $i$ based on the $\epsilon$-policy with the Q value $Q_i(o^i, u^i)$. To imply the importance of agents in the group, the importance weight $\lambda^i$ is obtained from Equation 13 to weight $Q_i(o^i, u^i)$ and form the weighted Q value $Q^i(o^i, u^i)$. Finally, the weighted Q values from all agents are aggregated to derive the global value $Q_{tot}$ using the QMIX mixing network:

$$Q_{tot}(o, u) = MixingNetwork(\lambda^1 \cdot Q_1(o^1, u^1), \cdots, \lambda^N \cdot Q_N(o^N, u^N)) \tag{14}$$

Subsequently, $Q_{tot}$ is trained to minimize:

$$\mathcal{L}_{tot} = \mathbb{E}_{\left\{o^i_t, o^i_{t+1}, u^i_t, u^i_{t+1}\right\}^N_{i=1}} [Q_{tot}(\{o^i_t, u^i_t\}^N_{i=1}) - y_t]^2 \tag{15}$$

where $y_t$ is calculated as follows, and $r_t$ is the reward at time step $t$:

$$y_t = r_t + \gamma \cdot [\widehat{MixingNetwork}(\hat{Q}^1(o^1_{t+1}, u^1_{t+1}), \cdots, \hat{Q}^N(o^N_{t+1}, u^N_{t+1}))] \tag{16}$$

To comply with the CTDE, the group controller and mixing network are only applied during the training process, while the local Q network, knowledge controller and knowledge integration are applied at both stages. The pseudo-code of our method is provided in Algorithm 1.

## 4 Experiments and analysis

In our experiments, we aim to answer the following questions: (1) Can the proposed framework improve the scalability of MARL algorithms? (2) What's the function of each module in our framework? (3) Is using suboptimal human knowledge justified, and how does its quality influence the learning process? To answer these questions, we conduct our experiments on challenging scenarios of the StarCraft Multi-Agent Challenge (SMAC) [24] with an increasing number of agents involved.

### 4.1 Experimental setting

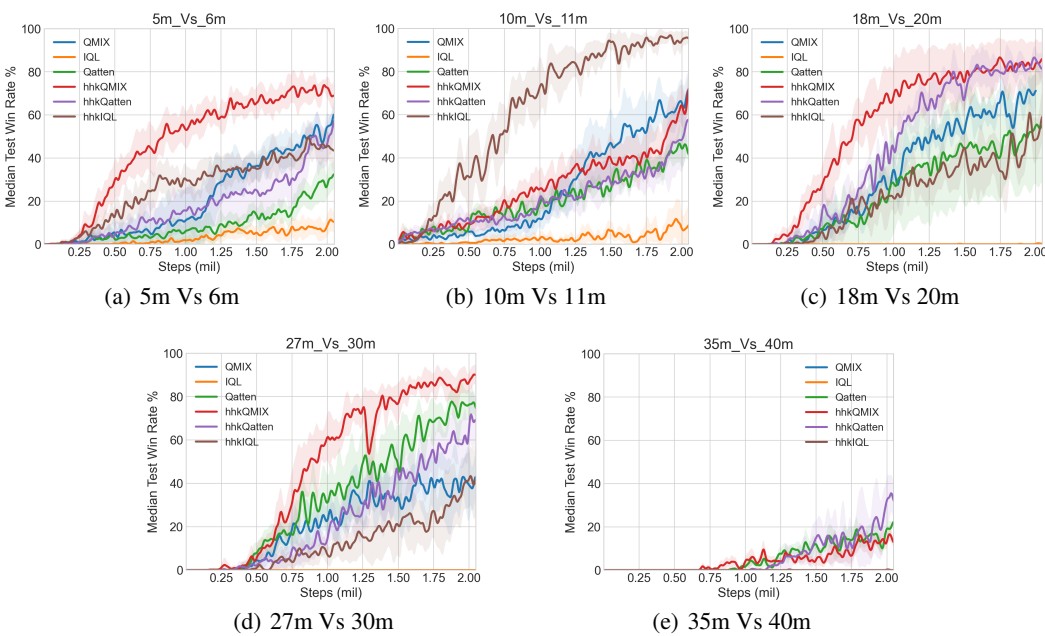

(a) 5m Vs 6m     (b) 10m Vs 11m     (c) 18m Vs 20m

(d) 27m Vs 30m     (e) 35m Vs 40m

Figure 4: Experimental results for our approaches and their corresponding baselines in five scenarios. The shaded region denotes standard deviation of average evaluation over 3 trials.

Focusing on micromanagement control, SMAC has been proposed as a common benchmark for MARL methods. In the following experiments, we test our framework on challenging scenarios in SMAC from the 'Hard' and 'Super Hard' categories, setting the game AI difficulty to 'Very Hard'. To demonstrate the impact of the agent number on algorithm performance, we add two self-designed scenarios with more agents involved. To verify the effect of our end-to-end framework, we combine our method with three famous MARL algorithms: IQL [30], QMIX [29], and Qatten [31], naming the corresponding approaches hhkIQL, hhkQMIX, and hhkQatten, respectively. In this study, each algorithm and its corresponding approach are configured with the same hyperparameters and neural network structures. Furthermore, all experimental results are derived across three separate trials with

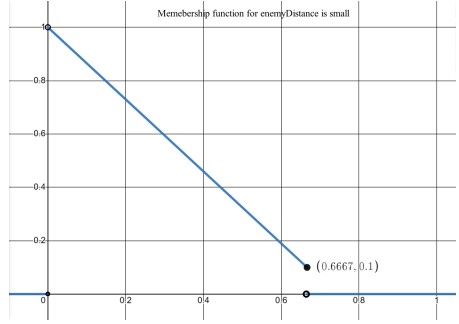

Figure 3: Membership function: '$e\_d$ is $small$'. X-axis denotes the observation value for variable $e\_d$ and Y-axis denotes the truth value.

different random seeds, with 32 test episodes in each trial. The shaded region in Figure 4 and 7, and the error bar in Figure 5 denote standard deviation in this work. Details of the hyperparameter settings are in Appendix A.5.

Instead of requiring high-quality expert demonstrations, this work aims to transfer suboptimal knowledge from humans, considering eight pieces of knowledge for challenging tasks in SMAC. Due to space limitations, we use the human prior knowledge: *'Attack the closest enemy'* as an example. As illustrated in Section 3.1, this abstract knowledge can be represented with a fuzzy logic rule:

- IF $e\_d$ is $small$, THEN $action$ is $attackEnemyId$.

Here, $e\_d$ represents the agent's observation of the distance between itself and enemies. The corresponding membership function for the fuzzy set $small$ is elaborated in Figure 3 and is defined as a simple linear function. More details about the applied human suboptimal knowledge can be found in Appendix A.6. As demonstrated, these fuzzy logic rules are abstract, suboptimal, and designed for specific states (e.g., attack actions are not always available), resulting in a win rate of 0% when

agents are solely manipulated by the proposed knowledge. Nonetheless, these rules are interpretable and easy to design and understand from the human aspect. All rules are designed based on the agent's local observation. Furthermore, it is worth mentioning that the proposed human prior knowledge is not well-crafted and is highly subjective, with no strict requirement for its appropriateness. Anyone can develop their fuzzy logic rules based on their own experience.

## 4.2 Result and evaluation

To answer the first question, we deploy the approaches in scenarios with an increasing number of agents. The learning curves for all approaches across all tasks are illustrated in Figure 4, and the performance comparison based on the number of agents is elaborated in Figure 5. In conclusion, the experimental results indicate that even based on highly abstract and low-performance human prior knowledge, our knowledge-guided approaches not only significantly accelerate the training process but also enhance the final performance across scenarios with various numbers of agents.

As the number of agents increases, the joint action-state space expands exponentially, making it exceedingly challenging for agents to learn from scratch. Despite these challenges, our approaches still manage to overcome the 'start-up' issue (Figure 4). By alleviating learning difficulties, while baseline algorithms fail in tasks with many agents (IQL in Figure 4(d), IQL and QMIX in Figure 4(e)), our approaches help agents overcome the initial learning challenges, improving scalability. Furthermore, as demonstrated by the win rate curves in Figure 4(a), our approaches greatly accelerate the training process, benefiting the initial stages of agent learning. Notably, our method (hhkIQL), even combined with the least effective baseline algorithm (IQL), achieves performance comparable to the best baseline algorithm (QMIX).

To better visualize our method in scenarios with different numbers of agents, we summarize the algorithms' performance in Figure 5. By leveraging human prior knowledge, our end-to-end method consistently helps MARL algorithms improve their performance across various numbers of agents. Additionally, it is worth mentioning the improvement our method brings to the IQL algorithm, shown in the first image of Figure 5. As one of the oldest fully decentralized MARL algorithms, IQL inherently struggles to handle coordination in large-scale MAS, displaying low performance in

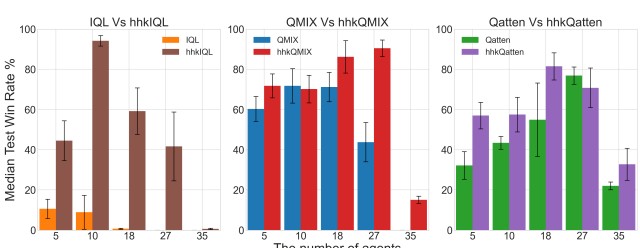

Figure 5: Performance comparison between baselines and our methods under the number of agents increase. The error bar is based on standard deviation

all scenarios. By contrast, benefiting from the integration of human knowledge and the relationship graph to mitigate 'start-up' issues and enhance cooperation, our approach significantly improves the scalability of the IQL algorithm.

## 4.3 Ablation study

To answer the following two questions, we design two ablation experiments: one to identify the function of each module and another to assess the influence of human suboptimal knowledge. Considering our improvement on the IQL algorithm, we use it as the basis in the '10m vs 11m' scenario to provide a clear comparison. The ablation results are detailed in Figure 7.

### 4.3.1 Module function

The experiment results on module functions are depicted in Figure 7(a), where 'hhkIQL-graphOnly' represents our method with only the group controller, 'hhkIQL-humanOnly' indicates our method integrating solely human knowledge, and 'hhkIQL-fixedKnow' denotes our method without trainable parameters in fuzzy logic rules.

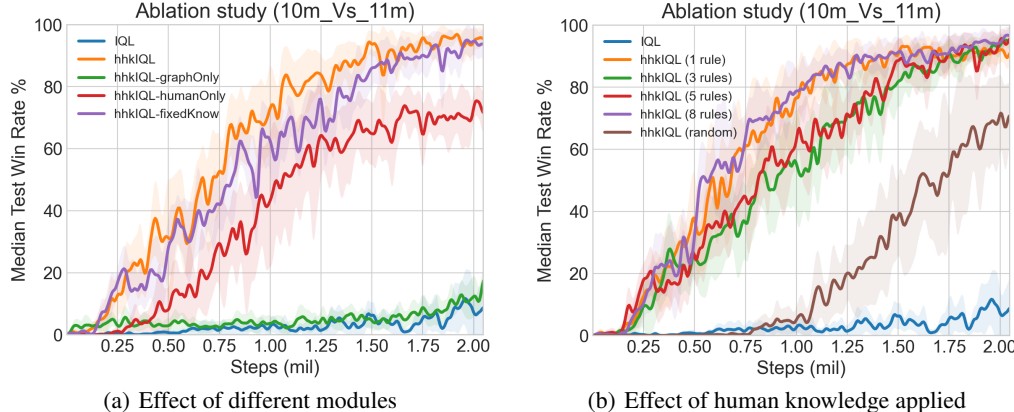

(a) Effect of different modules     (b) Effect of human knowledge applied

Figure 7: Ablation studies under '10m vs 11m' scenario. (a) ablation study on the function of each module in our method; (b) ablation study on the influence of various suboptimal human knowledge.

**Effect of group controller:** To elaborate on the role of the group controller, we first present agents' cooperation graph in Figure 6. The relationship graph reveals that the cooperation demand changes over time. With the group controller, agents can independently choose allies to collaborate with, and important agents are emphasized. As depicted in Figure 7(a), the group controller enhances cooperation among agents and accelerates the training process, evidenced by a higher convergence rate compared to that of the IQL algorithm.

**Effect of human knowledge:** Although the transferred knowledge is suboptimal, human prior knowledge can significantly improve performance. The knowledge integration, functioning as a 'CPG', allows human to effortlessly provide abstract knowledge from a high-level perspective. As a result in Figure 7(a), the knowledge-guided algorithm achieves overall better performance than the baseline algorithm. However, it is worth noting that this ablation algorithm can be further improved with the group controller installed, as evidenced by comparing it to the hhkIQL approach.

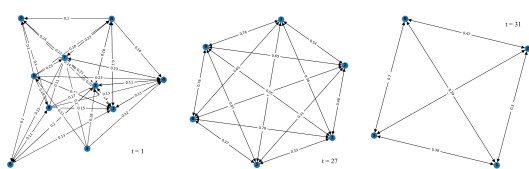

Figure 6: After the training of hhkIQL in '10m vs 11m' scenario, the change of each agent's $\lambda$ during a battle episode. The nodes are agents and the edges are agents' tendency to cooperate with others. $t$ is time step. The died agents are not shown in the graph (full details are in Appendix A.7).

**Trainable knowledge:** The benefits of trainable knowledge are demonstrated in Figure 7(a). Although both approaches achieve similar performance, the approach with trainable fuzzy logic rules realizes a faster convergence rate than its ablation counterpart. With the trainable knowledge controller, the prior knowledge is constantly optimized during training, enhancing the adaptation of the provided knowledge to the current task. As a result, agents can make better use of proposed knowledge and learn faster.

### 4.3.2  Human suboptimal knowledge influence

In this section, we explore the impact of suboptimal human knowledge and how our approach addresses inappropriate knowledge. The ablation results are described in Figure 7(b). To assess the effect of knowledge quality, we consider human knowledge with more fuzzy logic rules to be more comprehensive. For fairness, these rules are inherited among ablation approaches. For example, 'hhkIQL (8 rules)' will include the 5 rules used in 'hhkIQL (5 rules)', and so forth. Since knowledge from humans is highly subjective, it is hard to judge whether the transferred knowledge is inappropriate. To identify how our approach deals with inappropriate knowledge, we substitute the values in human preference vectors $Q_F^i$ with random value to represent inappropriate knowledge, which is denoted as 'hhkIQL (random)' in Figure 7(b).

As demonstrated in Figure 7(b), more comprehensive human knowledge can help agents achieve better performance. Although 'hhkIQL (1 rules)' can achieve faster learning speed than 'hhkIQL (3 rules)' and 'hhkIQL (5 rules)', it results in lower final performance. We guess that with fewer rules applied, there is a reduced learning burden on knowledge utilization, but this also leads to lower final outcomes. Notably, even though more comprehensive human knowledge is beneficial, the performance of these approaches remains similar. Furthermore, as the learning curve of 'hhkIQL (random)' indicates, the knowledge integration module can efficiently filter out negative knowledge. While it takes agents more time to learn how to utilize the proposed knowledge, 'hhkIQL (random)' can still outperform the baseline algorithm, highlighting the importance of the knowledge integration module. In conclusion, our approach does not rely on high-quality human knowledge, and the knowledge integration module can successfully mitigate the negative knowledge transfer problem, allowing humans to propose any knowledge they consider useful for agents.

## 5 Conclusion and future work

In this study, we introduce a novel hierarchical learning framework for enhancing coordination in large-scale MAS by leveraging suboptimal human knowledge. This framework consists of the group controller, the knowledge controller, and the knowledge integration, allowing humans to provide knowledge at the top level while agents develop their own policies at the bottom. Evaluated in SMAC with three famous algorithms, our end-to-end methods surpass corresponding baselines in learning speed and final performance, even with low-performance human knowledge integrated. Furthermore, this framework successfully improves the scalability of algorithm, handling scenarios with numerous agents where standard MARL algorithms fail. In the future, we will apply our approach in domains with even more agents involved, and explore its application to heterogeneous agents.

## Acknowledgment

This work was supported in part by the Ministry of Education, Culture, Sports, Science and Technology-Japan, Grant–in–Aid for Scientific Research under grant #JP24H00741, and part by the commissioned research by National Institute of Information and Communications Technology (NICT), JAPAN.

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

# A Appendix / supplemental material

## A.1 Fuzzy logic

Since human knowledge is highly abstract and uncertain, it is inappropriate to use hard rules to represent such prior knowledge [26]. Different from crisp sets, fuzzy logic, based on fuzzy set theory, can apply partial membership functions to represent fuzzy knowledge [32]. For a fuzzy set $F$, the $x$ in it can be described by a membership function $\mu_F(x)$ with range from 0 to 1, allowing the element partially belong to it:

$$\mu_F : X \longrightarrow [0, 1]$$

where $X$ refers to the universal set in a specific problem.

The fuzzy logic rule is usually in the form of 'IF $X$ is $A$ and $Y$ is $B$ THEN $Z$ is $C$'. Here, '$X$ is $A$' and '$Y$ is $B$' are called preconditions of the fuzzy rule, and '$Z$ is $C$' is the conclusion. The $X$, $Y$ and $Z$ are variables. And the $A$, $B$ and $C$ are fuzzy sets, also known as linguistic values. For each fuzzy set, it has a membership function $\mu_F$ to calculate the truth value $T$ of each precondition:

$$T_A = \mu_A(x_0) : X \to [0, 1], \quad T_B = \mu_B(y_0) : Y \to [0, 1]$$

where $x_0$ and $y_0$ are observation values for $X$ and $Y$, and $T_A$ and $T_B$ are truth values for preconditions '$X$ is $A$' and '$Y$ is $B$'. To get the conclusion of this fuzzy rule, it needs to satisfy both preconditions and the conjunction operator is applied:

$$\mu_{A \cap B}(x_0, y_0) = min(\mu_A(x_0), \mu_B(y_0))$$

Finally, we will get the conclusion's strength $\omega$, sometimes seen as the satisfaction level of the rule:

$$\omega = min(T_A, T_B) = min(\mu_A(x_0), \mu_B(y_0))$$

Summarizing, to abstract human prior knowledge with fuzzy logic rules, we need first to design the rules in the form of 'IF ... THEN ...' sentence. Then membership functions $\mu_F$ should be built for each preconditions to calculate their truth value $T$. Finally, the conjunction operator $min$ is applied to satisfy all the preconditions and get conclusion's strength $\omega$. Therefore, a fuzzy rule takes the observation values as input and outputs the value of conclusion to illustrate how likely to operate designed actions under current observation.

## A.2 Related work

Due to the expensive exploration, knowledge transfer has become an indispensable approach to enhance the scalability of MARL [11, 12]. On the one hand, the most straightforward implementation is to repurpose solutions from previous tasks obtained by agents [13]. On the other hand, various studies also emphasize the reuse of knowledge from auxiliary sources, such as human expert demonstrations [33].

As the "black box" approach is unsuitable for critical applications, the transfer method should be interpretable, prompting an increasing concern on Human-on-the-Loop [15]. By personally executing tasks, humans provide demonstrations for agents to record in state-action pairs which agents can mimic based on imitation learning [33, 34], inverse reinforcement learning [35, 36], and other human-focused methods [11, 37]. Unfortunately, these mainstream researches require step-by-step action demonstrations, heavily relying on high-quality and comprehensive expert demonstrations [16, 17].

While some efforts have aimed to mitigate the human burden, these solutions are generally limited to single- or two-player scenarios [20, 21, 38]. Fuzzy logic has been applied in previous work for knowledge representation [20], while their focus is on single-agent scenarios and the agent does not have self-policy development ability. As far as we know, the most successful work is from [27], who handle large-scale MAS with fuzzy agents. However, the use of human knowledge is not within their scope and their approach is more akin to agent knowledge transfer. Compared to previous works, our method, which can easily combine with various MARL algorithms, features a hierarchical learning scheme that human suboptimal knowledge is applied at top-level to enhance learning process of large-scale MAS. Based on the hyper-networks in knowledge integration, we are able to combine human preference with agent preference to empower agents with more knowledge selection freedom.

Our work also shares some similarities with the hierarchical RL methods [19, 23, 38, 39]. However, in contrast to these existing studies that pay more attention to decomposing challenging long-horizon tasks into simpler subtasks, our focus here is to connect humans and agents under a hierarchical structure for leveraging human knowledge and achieving more efficient learning in large-scale MAS.

### A.3 Symbol meaning

The meanings of symbols in this work is illustrated in Table 1

Table 1: Symbol meaning

| Symbol | Meaning |
|--------|---------|
| $s$ | global state |
| $r$ | reward |
| $D$ | replay buffer |
| $i$ | agent $i$ |
| $\{a_1, \ldots, a_N\}$ | all agents |
| $L$ | fuzzy logic rule $L$ |
| $M$ | fuzzy set |
| $\{u_1, \ldots, u_k\}$ | agent action space |
| $\{o_1, \ldots, o_m\}$ | agent observation space |
| $\{o_1, \ldots, o_z\}$ | observation values for fuzzy logic rule |
| $T$ | truth value of precondition |
| $\mu$ | membership function |
| $\omega$ | conclusion strength of fuzzy logic rule |
| $\beta$ | trainable weight of knowledge controller |
| $Q_F$ | human preference action value |
| $Q_{LOC}$ | agent preference action value |
| $Q_i$ | knowledge guided action value of agent $i$ |
| $\lambda_{i,j}$ | cooperation tendency of agent $i$ toward agent $j$ |
| $\lambda_i$ | agent $i$ cooperation tendency toward other agents |
| $\lambda^i$ | importance of agent $i$ in the group |
| $Q^i$ | $\lambda$ weighted action value of agent $i$ |
| $\alpha$ | parameter of knowledge integration hyper-network |
| $\theta$ | weight of integration module generated by integration hyper-network |
| $\Omega$ | hyperparameter of integration module |
| $Q_{tot}$ | global value from mixing network |
| $\mathcal{L}_{tot}$ | loss |
| $\gamma$ | discount factor |
| $h$ | history for RNN |
| $\tau$ | action observation history |
| $\epsilon$ | exploration rate |
| $\hat{\phantom{x}}$ | target network |

### A.4 Computational resource

In this work, we run our experiments in a computer with a CPU (13th Gen Intel Core i7-13700F 2.10 GHz), GPU (NVIDIA GeForce RTX 4080), and RAM (128GB). It takes us more than 550 GPU hours to finish all the experiments. It's worth mentioning that the '35m vs 40m' scenario is the most time-consuming experiment where a single run requires beyond 9 hours on average.

### A.5 Experiment hyperparameter

The hyperparameters for our experiments are shown in Table 2

### A.6 Suboptimal human knowledge applied in experiment

For challenging tasks in SMAC, the following 8 pieces of human knowledge are considered:

- Attack the closest enemy.
- Attack the enemy with the lowest HP.
- Get close to the closest enemy.
- Get close to the enemy with the lowest HP.

Table 2: Hyperparameters of experiment

| Parameter name | Value |
| --- | --- |
| Total timesteps | 2050000 |
| Number of environments | 8 |
| Number of test episodes | 32 |
| Test interval | 5000 |
| Update interval | 200 episodes |
| Optimizer | Adam |
| $\gamma$ | 0.99 |
| $\beta$ initialization | 1.0 |
| Batch size | 128 |
| Buffer size | 3000 |
| Learning rate | 0.001 |
| RNN layer hidden size | 64 |
| Group controller RNN hidden size | 64 |
| $\epsilon$ | $1.0 \to 0.05$ |
| Anneal time of $\epsilon$ | 50000 |
| QMIX mixing embed size | 32 |
| QMIX hypernet embed size | 64 |
| Qatten query embed size of layer 1 | 64 |
| Qatten query embed size of layer 2 | 32 |
| Qatten key embed size | 32 |
| Qatten head embed size of layer 1 | 64 |
| Qatten head embed size of layer 2 | 4 |
| Qatten attention head | 4 |
| Qatten number of constraint value | 32 |
| Knowledge integration hypernet size | 64 |
| Knowledge $\Omega$ | $1.0 \to 0.0$ |
| Anneal time of knowledge $\Omega$ | 1000 |

- Disperse when many agents are crowded together.

- Gather when there are few agents and they are far away.

- Get close to the ally who is attacking.

- Attack properly to avoid over-attacking.

The abstract knowledge can be represented with fuzzy logic rules as follows:

- IF $e\_d$ is $small$, THEN $action$ is $attackEnemyId$.

- IF $e\_hp$ is $small$, THEN $action$ is $attackEnemyId$.

- IF $e\_clo\_x$ is $PO$, THEN $action$ is $east$; IF $e\_clo\_x$ is $NE$, THEN $action$ is $west$; IF $e\_clo\_y$ is $PO$, THEN $action$ is $north$; IF $e\_clo\_y$ is $NE$, THEN $action$ is $south$.

- IF $e\_Lhp\_x$ is $PO$, THEN $action$ is $east$; IF $e\_Lhp\_x$ is $NE$, THEN $action$ is $west$; IF $e\_Lhp\_y$ is $PO$, THEN $action$ is $north$; IF $e\_Lhp\_y$ is $NE$, THEN $action$ is $south$.

- IF $n\_ally$ is $large$ AND $g\_ally\_d$ is $small$ AND $ally\_x$ is $PO$, THEN $action$ is $west$; IF ... AND $ally\_x$ is $NE$, THEN $action$ is $east$; IF ... AND $ally\_y$ is $PO$, THEN $action$ is $south$; IF ... AND $ally\_y$ is $NE$, THEN $action$ is $north$.

- IF $n\_ally$ is $small$ AND $g\_ally\_d$ is $large$ AND $ally\_x$ is $PO$, THEN $action$ is $east$; IF ... AND $ally\_x$ is $NE$, THEN $action$ is $west$; IF ... AND $ally\_y$ is $PO$, THEN $action$ is $north$; IF ... AND $ally\_y$ is $NE$, THEN $action$ is $south$.

- IF $ally\_attacking\_x$ is $PO$, THEN $action$ is $east$; IF $ally\_attacking\_x$ is $NE$, THEN $action$ is $west$; IF $ally\_attacking\_y$ is $PO$, THEN $action$ is $north$; IF $ally\_attacking\_y$ is $NE$, THEN $action$ is $south$.

- IF $n\_potential$ is $large$ AND $n\_attack$ is $proper$, THEN $action$ is $attackEnemyId$.

The membership functions for the fuzzy sets in each rule are elaborated in Figure 8.

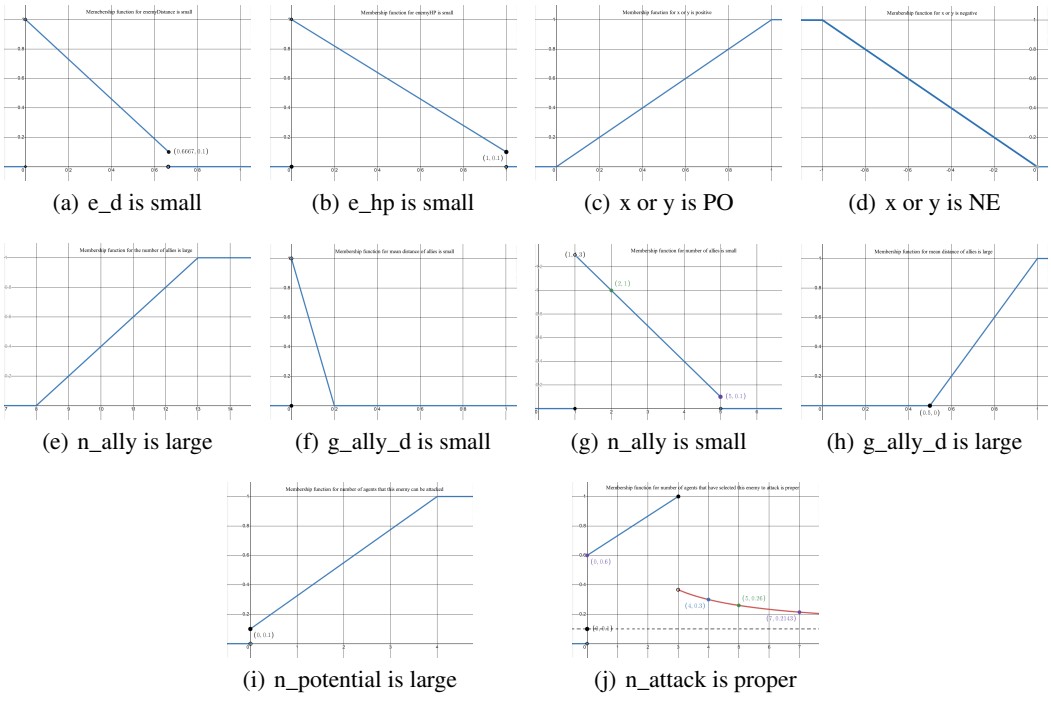

(a) e_d is small  (b) e_hp is small  (c) x or y is PO  (d) x or y is NE

(e) n_ally is large  (f) g_ally_d is small  (g) n_ally is small  (h) g_ally_d is large

(i) n_potential is large  (j) n_attack is proper

Figure 8: Membership functions used in SMAC.

## A.7 Dynamic graph

The full image of the dynamic graph based on group controller is elaborated in Figure 9.

## A.8 Limitations and broader impact

In this section, we will discuss the potential limitations of this work, which we aim to address in future research. First, as the proposed modules are shared among agents, we assume that the agents are homogeneous to alleviate the difficulty of knowledge design and computation complexity. However, exploring our approach with heterogeneous agents, which may require different kinds of knowledge, is an interesting direction. Second, even though fuzzy logic is a promising technique for knowledge abstraction, it is relatively primitive, and a better representation method is required to further improve performance, which is a consideration for future work. Third, in this work, we consider integrating suboptimal human knowledge to improve the performance of MARL algorithm and propose a hyper-network to avoid negative knowledge transfer. However, as illustrated in our ablation studies, more comprehensive knowledge should be beneficial. Therefore, discussing what kinds of knowledge are more appropriate and how to design effective knowledge is an interesting topic for future exploration. Finally, due to computational limitations, we only verify our approach in SMAC. Although we have applied ablation studies to enhance convincingness, it would be helpful to conduct experiments in other domains with more agents involved, which we plan for future work.

This work aims to contribute to the development of MARL algorithms. As with any field in machine learning, it is possible that improving the capabilities of these algorithms could lead to unethical uses. However, there are also many potential benefits to better cooperative AI, such as applications in disaster rescue robots among others. We believe that the potential benefits of developing more capable and cooperative AI outweigh the potential risks.

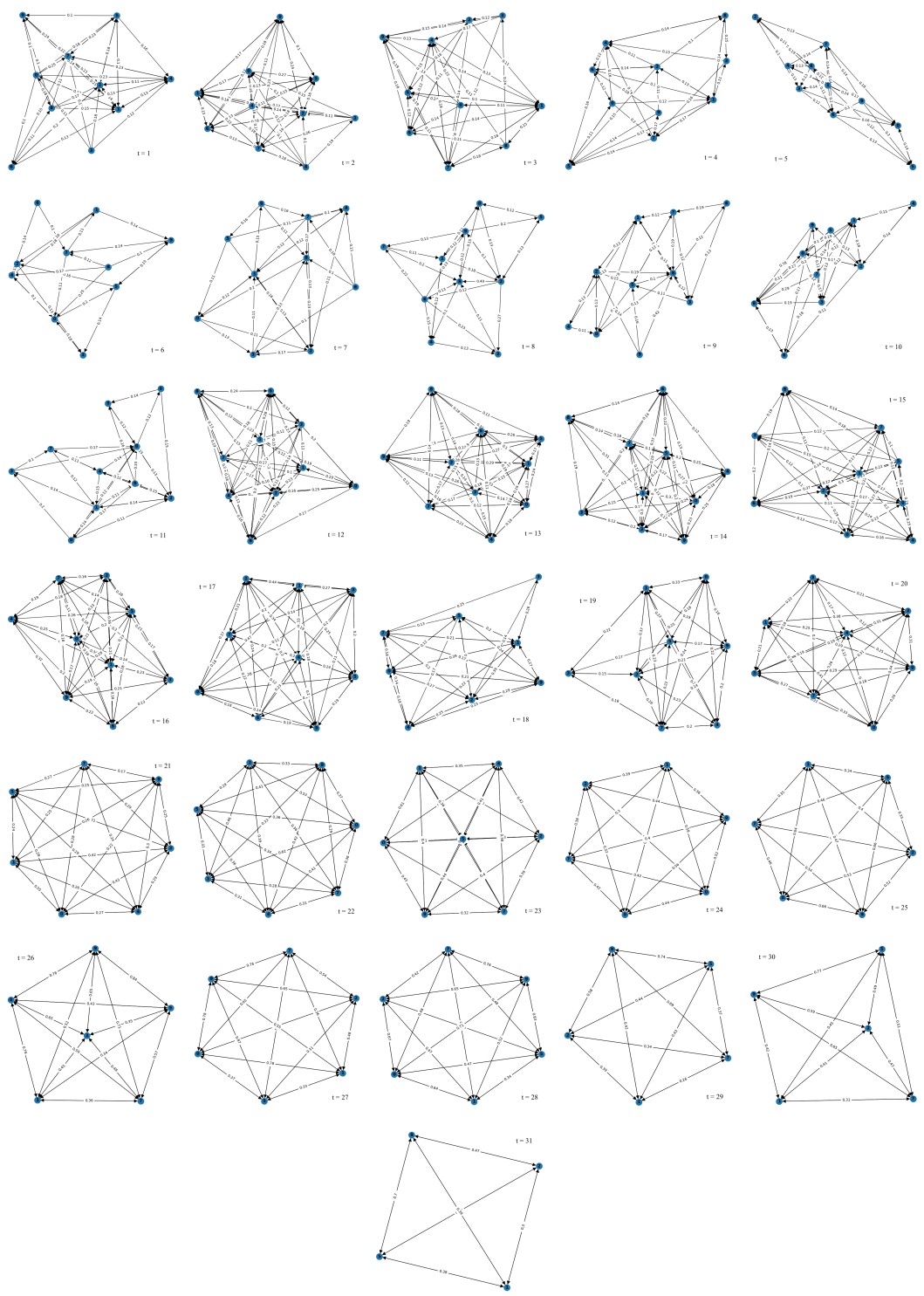

Figure 9: The cooperation graph from hhkIQL during one battle episode based on the change of each agent's $\lambda_i$ under '10m vs 11m' scenario.

