# OpenReview forum: "Integrating Suboptimal Human Knowledge with Hierarchical Reinforcement Learning for Large-Scale Multiagent Systems"
_NeurIPS.cc/2024/Conference — NeurIPS 2024 poster_

### Official Review · Reviewer_8Cur · 2024-07-12

**Soundness:** 2
**Presentation:** 3
**Contribution:** 3
**Rating:** 6
**Confidence:** 3

**Summary:**

The authors proposed a new framework (hhk-MARL) that integrates human abstract knowledge with hierarchical reinforcement learning to address the learning challenges in large-scale multi-agent systems. The framework employs fuzzy logic to represent human knowledge and a graph-based group controller to enhance agent coordination. Experimental results in the StarCraft Multi-agent Challenge demonstrate that the proposed approach significantly accelerates the training process and improves the final performance of the agents, even with imperfect human prior knowledge.

**Strengths:**

* Overall, the proposed hhk-MARL framework effectively integrates human knowledge with hierarchical reinforcement learning, providing a flexible and adaptive approach for multi-agent systems.

* Methodologically, the authors use fuzzy logic and hypernetworks to dynamically generate weights based on agent observations allowing for the seamless combination of human and agent preferences.

* In the experimental results, the comprehensive experiments demonstrate the framework's efficacy and robustness, showing significant improvements over baseline methods in various scenarios.

**Weaknesses:**

While the paper is well-written overall, it lacks detailed comparisons with prior research and thorough explanations of the human knowledge used. Specifically, section 3.1 does not clearly justify the choice of fuzzy logic over other approaches, which could also handle uncertainty and abstract knowledge effectively. Additionally, the paper can explain how its approach differs from previous works such as KoGuN and Shi 2021, to clarify its unique contributions. In section 3.2, the new aspects of using hypernetworks for knowledge integration should be more distinctly highlighted in comparison to similar methodologies. For the details, see the following Questions.

**Questions:**

1. The paper employs fuzzy logic in the hhk-MARL framework for integrating human knowledge, but it does not provide a clear explanation in section 3.1 for choosing fuzzy logic over other methods such as Bayesian networks, a combination of reinforcement learning and heuristics, or neural network-based knowledge representation. These alternative methods could also handle uncertainty and abstract knowledge effectively. A more detailed justification for selecting fuzzy logic would enhance the understanding of its advantages in this context.

2. The idea of incorporating fuzzy logic into hierarchical RL has been introduced in prior works such as KoGuN [20] and in Shi 2021 [a]. However, it is essential to explain the differences between these approaches and the proposed method in section 3.1. Highlighting these distinctions will clarify the unique contributions of this work and demonstrate how it advances beyond existing methodologies.

3. In section 3.2 on Knowledge Integration, it is important to highlight the novel aspects of the proposed method. The integration of human knowledge using hypernetworks to dynamically generate weights based on agent observations presents a unique approach. However, the methodology bears similarities to existing works such as HAMXCS [21] which also incorporates heuristic knowledge into reinforcement learning, Hierarchical RL for Self-Driving [19] which uses hierarchical reinforcement learning for decision-making, and KoGuN, which refines fuzzy logic rules with hypernetworks. A clearer distinction of the innovative elements in comparison to these related works would strengthen the contribution of this paper.

4. Appendix A.6 presents the suboptimal human knowledge rules, but it is unclear who determined these rules and to what extent they are incomplete or reasonable. Given that they are labeled as suboptimal, a clearer explanation of their limitations and adequacy is necessary, especially for readers unfamiliar with Starcraft. This information is crucial for understanding the potential applicability and adaptation of these rules to other tasks.

**Limitations:**

The authors have adequately addressed the limitations and potential negative societal impact of their work.

---

> ### Author Rebuttal · Authors · 2024-08-05
>
> Thank you very much for your kind review. We are pleased that you thought we have provided a flexible and adaptive approach for multi-agent systems.
>
> We have addressed your comments below. We hope that this clarifies any concerns that you had and strengthens your support for the paper.
>
> >Question1:...it does not provide a clear explanation in section 3.1 for choosing fuzzy logic...A more detailed justification for selecting fuzzy logic would enhance the understanding
>
> Thank you for your comment. We will revise Section 3.1 to further depict our motivation for selecting fuzzy logic. We consider fuzzy logic because it is closer to the human perceptions and knowledge structures. Compared to the mentioned alternative methods, fuzzy logic is more interpretable, giving humans more freedom and control over knowledge design and representation. Moreover, as the intermediary to control agents, using fuzzy logic can reduce complexity, making it more suitable for training large-scale multi-agent systems. Furthermore, the framework can benefit from the advantage of fuzzy logic in generalization. Since previous research has revealed these advantages of fuzzy logic on knowledge representation [20, 38], it motivated us to leverage fuzzy logic for human prior knowledge representation.
>
> We will revise as:
>
> *Section 3.1: “Compared to other knowledge representation methods, fuzzy logic is closer to the structure of human knowledge, making it more interpretable. Furthermore, it has been proven that fuzzy logic is more suitable for training large-scale multi-agent systems with the advantage of generalization [38]. Inspired by previous works on knowledge representation with fuzzy logic [20, 38], we leverage fuzzy logic to abstract human prior knowledge in this work. The general form…”*
>
> >Question2:The idea of incorporating fuzzy logic into hierarchical RL has been introduced in prior works such as KoGuN [20] and in Shi 2021[a]. it is essential to explain the differences between these approaches and the proposed method in section 3.1...
>
> Thank you for your comment. The comparison between our work and previous research is proposed in Appendix A.2. We will further clarify the difference in the related work section. KoGuN [20] applies fuzzy logic in the single-agent scenario where the agent only learns how to leverage human knowledge without self-policy development ability. In Shi 2021 [13], the authors consider an all-purpose cross-task transfer that transfers knowledge among agents based on features extracted from neural networks. Different from these two works, we leverage fuzzy logic to connect agents and humans in multi-agent systems.
>
> We will revise as:
>
> *Appendix A.2: “…of MARL [11, 12]. On the one hand, the most straightforward implementation is to repurpose solutions from previous tasks obtained by agents [13]. On the other hand, various studies also emphasize the reuse of knowledge from auxiliary sources, such as human expert demonstrations [32]. ……Fuzzy logic has been applied in previous work for knowledge representation [20], while their focus is on single-agent scenarios and the agent does not have self-policy development ability. As far as we…”*
>
> >Question3:In section 3.2 on Knowledge Integration, it is important to highlight the novel aspects...
>
> Thank you for your comment. Some discussion about knowledge transfer methods is given in Appendix A.2 to exhibit the novelty of our approach. We will detail the motivation for applying hyper-networks in Section 3.2 and further clarify the distinction in Appendix A.2. In general, through the hyper-networks based Knowledge Integration module, agents can still maintain learning ability from the local Q network, and the human prior knowledge is not distorted. In comparison, KoGuN [20] requires the global state information to refine the compressed human prior knowledge, and the agent action preference is not considered. HAMXCS [21] also requires global information for the two-player competitive game, and a neural network is applied to construct an opponent model. In the approach for self-driving [19], their focus is more on decomposing challenging long-horizon tasks into simpler subtasks. Although it mitigates the reliance on labelled driving data, the human demonstration still needs to be step-by-step samples.
>
> We will revise as:
>
> *Section 3.2: “Although applying a concatenated neural network as the knowledge integration is straightforward, it is difficult to capture the dynamic knowledge requirements in different states. To allow agents to automatically adapt to human guidance, motivated by previous research [20], we propose a hyper-networks based knowledge integration that allows agents to refine the proposed prior knowledge based on the local observation. As shown in Figure 2… ”; Appendix A.2: “…large-scale MAS. Based on the hyper-networks in knowledge integration, we are able to combine human preference with agent preference to empower agents with more knowledge selection freedom.”*
>
> >Question4:...it is unclear who determined these rules and to what extent they are incomplete...explanation of their limitations and adequacy...
>
> Thank you for your comment. In this work, these human knowledge rules are specifically designed for SMAC, and the design of knowledge is correlated with the applied domain. As declared in Section 4.1, the proposed knowledge for SMAC is suboptimal, resulting in a 0% win rate when agents are solely manipulated by the proposed knowledge. Furthermore, our approach does not rely heavily on the selection of the rules, and we give users the freedom to define these fuzzy logic rules based on their domain knowledge. As shown in our ablation study (Section 4.3.2), agents can selectively adapt to prior knowledge through the Knowledge Integration module, allowing humans to propose any knowledge they consider useful for agents. In the future, we will investigate which kinds of knowledge are more appropriate and how to design effective knowledge.

---

> ### Author Response · Authors · 2024-08-11
> **Seek open dialogue**
>
> We are grateful to your earlier constructive comments, and we hope our rebuttal has addressed the questions raised. If you need further clarifications, we are very happy to follow up to improve our final version to meet the high standard of this conference.

---

> > ### Comment · Reviewer_8Cur · 2024-08-12
> > **Thank you for the rebuttals**
> >
> > Thank you for the detailed responses and clarifications. These revisions will certainly enhance the clarity and understanding of your paper. I have no further concerns at this time.

---

> > > ### Author Response · Authors · 2024-08-12
> > > **Thank you for your response**
> > >
> > > Thank you very much for taking the time to respond to our rebuttal and your effort in engaging with us. We are glad that we were able to address your concerns.

---

### Official Review · Reviewer_kCt8 · 2024-07-12

**Soundness:** 3
**Presentation:** 3
**Contribution:** 3
**Rating:** 6
**Confidence:** 4

**Summary:**

This paper integrates an human in the loop to provide knowledge that improves learning in marl. This is done through a hierarchical structure, but ultimately it is up to the agents the final decision of accepting the human suggestions (hierarchy comes from human knowledge to agents). Overall, there is an integration of human knowledge with what the agents learn.

**Strengths:**

The paper is well organized and it is interesting. The idea of integrating human knowledge is this kind of tasks is interesting and it makes sense. Mostly, the paper is easy to follow and understand, with some exception that I outline below.

**Weaknesses:**

While I find this paper interesting, I have some remarks, as noted below:
* in line 131 it is stated that $M_L^I$ corresponds to a fuzzy set; however, it is not clear what $M$ is exactly, and how it relates to fuzzy logic in this context; it is not easy to understand how the relation between $O$ and $M$ is calculated and how is the membership function $\mu$ implemented
* from my understanding, when integrating the proposed approach with IQL, there is not a mixer as in the other value function factorization methods like QMIX; this leads me to think that the lack of a mixer could be the reason behind the huge improvements seen in Figure 4 for IQL when combined with this method; can it mean that the mixer can be slightly detrimental in the overall framework?

**Questions:**

In addition to the points above, I have some specific questions that I would like the authors to comment on:
* what are the limitations of considering only a small set of human opinions, like the 8 ones defined in appendix 6?
* if a certain human knowledge concerns only one specific agent, is it still given to the others as well? if yes, how does it affect learning? do the others get confused by that knowledge?
* how do the agents decide if they accept a certain human suggestion? is it a joint decision? if yes, can they reason accurately if the decision concerns a specific agent, but has nothing to do with the others?
* is the knowledge controller deterministic? or was it trained a priori with human knowledge?
* it is stated in lines 179-180 that the group controller is used only during training to not violate CTDE; what about the knowledge integration, is it also used only during training? or also during execution? since it receives human knowledge from all agents too
* the knowledge integration module seems to follow an interesting approach, since the weights of the second network are generated by an initial network based on the observations of the agents given as inputs; could the authors elaborate on the motivations for this approach? why are the weights generated from a first network and not only using one network?
* i can see the authors focused in environments involving marines only; is there a specific reason for that? is it because of the availability of prior human knowledge? or how hard it would be to create a knowledge controller network for other cases?

---

> ### Author Rebuttal · Authors · 2024-08-05
>
> We would like to thank you for your review and your kind words about our paper. We are pleased that you found our idea of integrating human knowledge to be interesting.
>
> Below we have addressed your questions. We hope that this strengthens your support for the paper.
>
> >Weakness1: ...what $M$ is exactly, ...$\mu$ implemented
>
> Thank you for your comment. The fuzzy set $M$ and membership function $\mu$ are the components of fuzzy logic rules that are designed by humans. $M$ denotes the fuzzy set and $\mu$ represents the relationship between $O$ and $M$ as follows: $\mu_M(o):O\rightarrow [0,1]$. An example of the fuzzy logic rule is proposed in Section 2.3: ’IF $O$ is *high*, THEN *action* is *read*’ regarding the knowledge of ’Read paper with high citation score’. Here, $O$ is the observation of the citation score, and *high* is a fuzzy set $M$ whose membership function could simply be, $\mu_{high}(o): clip[0.05 \cdot o,0,1]$. It is worth mentioning that we give the users freedom to define these fuzzy logic rules and our approach does not rely heavily on the selection of the rules. We are deeply sorry that our Section 2 and Table 1 may not be clear enough, and we will modify the structure of our paper to further clarify it.
>
> >Weakness2:...can it mean that the mixer can be slightly detrimental...
>
> Thank you for your comment. From our understanding, mixers can influence the cooperation among agents and different types of mixers may have diverse benefits, as even baseline algorithms exhibit different performances in different scenarios. Still, as shown in Figure 4, our framework can easily be combined with various MARL algorithms and enhance their performance. As in this work we just consider a general approach for MARL algorithms, we do not further discuss the influence of mixer and answer the question of what kinds of mixers are more suitable. We will investigate this in our future work.
>
> >Question1:What are the limitations of considering only a small set of human opinions...
>
> Thank you for your comment. In this work, those human opinions are specifically designed for SMAC, and the design of knowledge is correlated with the applied domain. As proven by imitation learning and inverse reinforcement learning [33, 35], more comprehensive guidance should be more beneficial. As shown in our ablation study (Section 4.3.2), more comprehensive knowledge can improve the learning speed and final performance. However, because of the complexity of multi-agent systems, it is challenging to propose the overall demonstrations, while our approach allows the use of suboptimal human guidance. Even if a small set of human opinions is proposed, the Knowledge Integration module can allow agents to selectively adapt to the prior knowledge while maintaining self-learning ability. Based on the domain, if it is possible, then a larger set of human opinions can be more instructive, while a small set is still acceptable for our framework.
>
> >Question2:If a certain human knowledge concerns only one specific agent ... do the others get confused by that knowledge?
>
> Thank you for your comment. In this work, we focus on the homogeneous agents where agents share similar goals, observations, etc. The proposed human prior knowledge is shared among all agents. As agents can selectively adapt to the proposed human knowledge based on the Knowledge Integration module, it should not affect the learning and other agents should not get confused.
>
> >Question3:How do the agents decide if they accept a certain human suggestion? is it a joint decision?...
>
> Thank you for your comment. Agents use local observation to decide the utilization of the proposed human suggestions. This is not a joint decision, as each agent can use its Knowledge Integration to decide whether to accept a certain suggestion.
>
> >Question4:Is the knowledge controller deterministic...
>
> Thank you for your comment. The Knowledge Controller module is deterministic and is set up by humans before the training and then adjusted by agents through the reinforcement learning process.
>
> >Question5:...about the knowledge integration, is it also used only during training? or also during execution...
>
> Thank you for your comment. The Knowledge Integration module is applied during both training and execution. Each agent selectively adapts to the proposed knowledge based on its Knowledge Integration, which is shared among agents.  We will emphasize this in our paper.
>
> >Question6:...could the authors elaborate on the motivations for hyper-network...
>
> Thank you for your comment. The hyper-network structure can offer more advantages than a concatenated neural network. Even though using a neural network is straightforward, it is hard to capture the dynamic knowledge requirements in different states. Moreover, using a feed-forward network to generate weights for another network, hyper-networks are more in line with the semantics of the Knowledge Integration module. It allows agents to selectively adapt to human guidance through local observation, which is hard to achieve in a single neural network. As hyper-networks have been proven to be more beneficial for knowledge refining in previous work [20], we apply such a structure in this work motivated by its advantages. We will emphasize the importance of hyper-networks in our paper.
>
> >Question7:...availability of prior human knowledge? or how hard it would be to create a knowledge controller...
>
> Thank you for your comment. In this work, we focus on the homogeneous agent setting. As the marine is the most common unit in SMAC, we deploy our approach in scenarios with marines involved. It is worth mentioning that our method is not limited to a single scenario. To reduce human burden, we allow the proposed human knowledge to be suboptimal and give users the freedom to design the transferred knowledge. As empirical results show (Section 4.1 and Figure 6), our Knowledge Integration module can greatly reduce the knowledge design requirements.

---

> ### Author Response · Authors · 2024-08-11
> **Seek open dialogue**
>
> We are grateful to your earlier constructive comments, and we hope our rebuttal has addressed the questions raised. If you need further clarifications, we are very happy to follow up to improve our final version to meet the high standard of this conference.

---

### Official Review · Reviewer_JxHm · 2024-07-19

**Soundness:** 3
**Presentation:** 3
**Contribution:** 3
**Rating:** 6
**Confidence:** 2

**Summary:**

In this paper, the authors propose a novel method to tackle the multi-agent reinforcement learning problem. They do so by combining human abstract knowledge with hierarchical reinforcement learning. Specifically, human knowledge in the form of fuzzy logic rules is combined, at the top level, with each individual agent’s decisions, learned at the bottom level. Then, a graph-based group controller performs agent coordination to decide what the action at each step should be. The authors evaluate the proposed method on the StarCraft multi-agent Challenge, combined with three algorithms (IQL, QMIX, and Qatten). The results indicate that the proposed approach is capable of improving the overall performance.

**Strengths:**

The authors tackle the multi-agent reinforcement learning problem in a creative and novel way, combining human feedback with hierarchical techniques. Moreover, it is a very interesting idea to provide a general algorithm that can be coupled with any existing MARL technique. In this way, it is possible to take advantage of the benefits of previously proposed algorithms while also incorporating new ideas to improve overall performance.

The paper is well-written, and all the high-level ideas behind the different components are well-explained.

**Weaknesses:**

The empirical evaluation is thorough, and it indicates that the claims are correct. However, only three samples are rarely enough to draw strong conclusions.

I believe some crucial parts of the algorithm were not very detailed. In particular, what are the learning rules/loss functions for each of the components? That is, how is $\beta$ trained? How is the knowledge integration component trained? With the current description, I believe it would be very hard for someone to replicate the method.

**Questions:**

For the ablation studies of the impact of each component on the final performance, it would also be interesting to see how much of the agent’s Q is used vs how much of the human knowledge Q is used for the final Q prediction. That is, what is the magnitude of the impact that the human knowledge component has over the Q predictions?

**Limitations:**

The authors discuss the limitations of the work. No major concerns.

---

> ### Author Rebuttal · Authors · 2024-08-05
>
> Thank you very much for your review. We are pleased that you found our method to be creative and novel and that you thought we had set a general approach for MARL algorithms.
> We address your concerns below and hope this will strengthen your support for the paper.
>
> > The empirical evaluation is thorough, and it indicates that the claims are correct. However, only three samples are rarely enough to draw strong conclusions.
>
> Thank you for this comment. In our experiments, our results are based on three separate runs, with the number of the test episodes during each run being 32 (shown in Table 2). Therefore, the number of samples to draw the conclusions is actually 96. The reason we deploy three different trials is to avoid random initialization deviation, which is a common strategy in previous works [Böhmer et al., 2020 (4 trials); Zhou et al., 2022 (4 trials); Zhong et al., 2024 (3 trials)]. As the standard deviation shown in Figure 5 is relatively small, it confirms the consistency of our results. We believe such results are sufficiently strong to draw a conclusion. We will clarify it in our paper to avoid this misunderstanding.
>
> We will revise the paper as follows:
>
> *Section 4.1: “Furthermore, all experimental results are derived across three separate trials with different random seeds, with 32 test episodes in each trial. The shaded region…”*
>
> * Böhmer, W., Kurin, V., & Whiteson, S. (2020, November). Deep coordination graphs. In International Conference on Machine Learning (pp. 980-991). PMLR.
> * Zhou, H., Lan, T., & Aggarwal, V. (2022). Pac: Assisted value factorization with counterfactual predictions in multi-agent reinforcement learning. Advances in Neural Information Processing Systems, 35, 15757-15769.
> * Zhong, Y., Kuba, J. G., Feng, X., Hu, S., Ji, J., & Yang, Y. (2024). Heterogeneous-agent reinforcement learning. Journal of Machine Learning Research, 25(1-67), 1.
>
> > I believe some crucial parts of the algorithm were not very detailed. In particular, what are the learning rules/loss functions for each of the components? That is, how is $\beta$ trained? How is the knowledge integration component trained? With the current description, I believe it would be very hard for someone to replicate the method.
>
> Thank you for your comment. We will add some details to the paper to further clarify these questions (in Section 3.1, 3.2, 3.4, and Algorithm 1). In general, all the components (including $\beta$ and Knowledge Integration) follow the traditional Q learning process, and the overall loss function is proposed in Equation 15:
>
> $\mathcal{L} _{tot} = \mathbb{E} _{[ o_t^i,o _{t+1}^i,u_t^i,u _{t+1}^i ] _{i=1}^N }[Q _{tot}([ o_t^i,u_t^i ] _{i=1}^N)-y_t]^2$
>
> 1. For $\beta$, they are initialized to 1 and backpropagated from $Q_F$ in the Knowledge Controller, which is similar to a neural network.
> 2. For the Knowledge Integration, the reward signal is backpropagated from the $Q_i$ to update the parameters of the integration $k_θ (\cdot)$ and then further update the parameters of the hyper-network $h_α (\cdot)$.
>
> We will revise the paper as follows:
>
> *Section 3.4: “This learning framework is end-to-end and can be combined with various MARL algorithms where the training of the knowledge controller, knowledge integration, and group controller module is based on the traditional Q learning process. To clarify this process…”*
>
> *Section 3.1: “These trainable weights are initialized at 1 to avoid disturbing the prior knowledge, and then adjusted through the reinforcement learning based on the reward signal. However, it…”*
>
> *Section 3.2: “…in knowledge adjustment. Similar to the knowledge controller, the knowledge integration is also trained based on the reinforcement learning process and this module is also shared among all agents. ”*
>
> > For the ablation studies of the impact of each component on the final performance, it would also be interesting to see how much of the agent’s Q is used vs how much of the human knowledge Q is used for the final Q prediction. That is, what is the magnitude of the impact that the human knowledge component has over the Q predictions?
>
> Thank you for noticing this. This is a very interesting question that is strongly related to designing beneficial human prior knowledge. To evaluate this, we can extract the weight of hyper-networks in the Knowledge Integration module to reveal the components of these two parts. As shown in our second ablation study (Figure 6(b)), the inappropriate knowledge will be automatically filtered out by agents, which may partially answer this question. Since our focus here is on connecting humans and agents in a hierarchical structure to boost the learning process, we do not consider this human knowledge representation aspect. We will address this question in our future work to guide users on how to design more advantageous human prior knowledge for better behavior guidance.

---

> > ### Comment · Reviewer_JxHm · 2024-08-13
> >
> > Thank you for the detailed response, and I am sorry for the late reply. This definitely clarified my questions/misunderstandings and I will update my score accordingly.

---

> > > ### Author Response · Authors · 2024-08-13
> > > **Thank you for your response**
> > >
> > > Thank you very much for taking the time to respond to our rebuttal and for updating your score! We are glad that we were able to address your concerns. We will definitely incorporate the clarifications from the rebuttal to the revised version of the manuscript.

---

> ### Author Response · Authors · 2024-08-11
> **Seek open dialogue**
>
> We are grateful to your earlier constructive comments, and we hope our rebuttal has addressed the questions raised. If you need further clarifications, we are very happy to follow up to improve our final version to meet the high standard of this conference.

---

### Author Rebuttal · Authors · 2024-08-05

We sincerely thank the reviewers for their constructive comments. We are encouraged that all the reviewers think our paper is well organized and our approach to integrating human knowledge with multi-agent reinforcement learning is flexible and effective (Reviewer JxHm, kCt8, and 8Cur). It is a great honor that the reviewers find our idea interesting (Reviewer JxHm and kCt8). We are excited that you find our method to be novel (Reviewer JxHm) and technically sound (Reviewer kCt8 and 8Cur), with experiments that are thorough and comprehensive (Reviewer JxHm and 8Cur). We answer the comments from reviewers based on part 6 (weaknesses) and part 7 (questions), to address your concerns:

1. We further clarify the training process of our approach with more details added.
2. We further clarify our motivation for choosing fuzzy logic and using hyper-networks.
3. We further clarify the difference between our approach and previous works.
4. We further clarify the technical details about designing and leveraging fuzzy logic rules.
5. We explain the reason for our experimental setting.
6. We answer the specific comments from each reviewer.

We will ensure that all the concerns are addressed in the revised paper. Our detailed response to the reviewers’ comments is shown below and hope this will strengthen your support for the paper.

---

### Decision · Program_Chairs · 2024-09-25

**Decision:**

Accept (poster)

**Comment:**

The paper explored an interesting (and increasingly relevant with LLMs) approach to leveraging human knowledge with hierarchical RL for multi-agent systems. All reviewers agreed that it was a technically solid contribution. There were a few clarifications requested by the reviewers which should help make the final version clearer.